# Metabolomics of a neonatal cohort from the Alliance for Maternal and Newborn Health Improvement biorepository: Effect of preanalytical variables on reference intervals

**Lena Jafri[1]\*, Aysha Habib Khan[2], Muhammad Ilyas[3], Imran Nisar[3], Javairia Khalid[3], Hafsa Majid[2], Aneeta Hotwani[3], Fyezah Jehan[3]\***

1 Department of Pathology and Laboratory Medicine, Chemical Pathology, Aga Khan University, Karachi, Pakistan, 2 Department of Pathology and Laboratory Medicine, Aga Khan University, Karachi, Pakistan, 3 Department of Pediatrics and Child Health, Aga Khan University, Karachi, Pakistan

\* lena.jafri@aku.edu (LJ); fyezah.jehan@aku.edu (FJ)

## Abstract

### Background

The study was conducted to determine reference interval (RI) and evaluate the effect of pre-analytical variables on Dried blood spot (DBS)-amino acids, acylcarnitines and succinylace-tone of neonates.

### Methodology

DBS samples were collected within 48–72 hours of life. Samples were analyzed for bio-chemical markers on tandem mass spectrometer at the University of Iowa. Comparison of RI across various categorical variables were performed.

### Results

A total of 610 reference samples were selected based on exclusion criteria; 53.2% being females. Mean gestational age (GA) of mothers at the time of delivery was 38.7±1.6 weeks; 24.5% neonates were of low birth weight and 14.3% were preterm. Out of the total 610 neo-nates, 23.1% were small for GA. Reference intervals were generated for eleven amino acids, thirty-two acylcarnitines and succinylacetone concentrations. Markers were evalu-ated with respect to the influence of gender, GA, weight and time of sampling and statisti-cally significant minimal differences were observed for some biomarkers.

### Conclusion

RI for amino acids, succinylacetone and acylcarnitine on DBS has been established for healthy neonates, which could be of use in the clinical practice. Clinically significant effect of GA, weight, gender and time of sampling on these markers were not identified.

**Data Availability Statement:** The datasets generated and/or analysed during the current study are now available as coded data without any

identifiers on FigShare; repository link: https://figshare.com/articles/journal_contribution/AAACARN_data_for_registry_xlsx/21081391.

**Funding:** This work was supported by the Bill & Melinda Gates Foundation.

**Competing interests:** The authors have declared that no competing interests exist.

**Abbreviations:** NBS, Newborn screening; IMDs, Inherited Metabolic Disorders; DBS, dried blood spot; AKU, Aga Khan University; RI, reference interval; AMANHI, Alliance for Maternal and Newborn Health Improvement; CLSI, Clinical Laboratory Standards Institute; CV, coefficient of variation; SD, standard deviations; CI, Confidence intervals; AGA, appropriate for gestational age; SGA, small for gestational age; WHO, world health organization; GA, gestational age; LBW, low birth weight; LC-MS/MS, liquid chromatography with tandem mass spectrometry.

## Introduction

Newborn screening (NBS) is a well-established public health program in most high-income countries to identify asymptomatic neonates with disorders for which prompt treatment by avoidance of fasting, special diets, cofactor and/or vitamin supplementation may decrease the risk of morbidity and mortality [1–3]. At present, newborns are being screened or diagnosed for more than thirty Inherited Metabolic Disorders (IMDs) using tandem mass spectrometry in many developed countries with expanded newborn screening. Expanded newborn screening includes profiling of amino acids, acylcarnitines and succinylacetone for screening of a large group of IMDs including aminoacidopathies and organic acidemias [4–6]. In Pakistan, a national NBS program does not exist and dried blood spot (DBS) testing is still not the standard of care [7–9] but isolated efforts from private sector are being made. With a high incidence, approximately 46–62%, of consanguineous marriages in Pakistan it is presumed that there is a high incidence of IMDs [10–13]. A previous report from the Aga Khan University (AKU) Pakistan has shown prevalence rates of 4.7% from our center in 2016 in high-risk children [10]. Recent screening of about 22,500 high risk individuals has shown this to increase to 7.5% (unpublished data). One of the challenges in Pakistan is the non-availability of country specific reference interval (RI) for screening newborns for IMDs using DBS.

RI is the interval between two limiting values within which 95% of the results for apparently healthy individuals would fall usually between the 0.025 and 0.975 fractiles of the distribution of test results for the healthy or reference population [14]. Ascertaining RI in neonates and children is challenging due to heterogeneity of phenotypic and epidemiological variables such as gender, ethnicity, gestational age, weight and environmental conditions [15–18]. The RI and subsequently determined cutoffs may vary between NBS programs from one region to another [14]. It is therefore recommended and essential for screening laboratories to establish and verify population specific RI and cut-offs for each disease marker analyzed.

The objective of the current study was to determine RI for amino acids (eleven amino acids and seven ratios), acylcarnitines (thirty-two acylcarnitines and fourteen ratios) and succinylacetone on DBS from a cohort of neonates enrolled in the AMANHI (Alliance for Maternal and Newborn Health Improvement) bio repository study at AKU Pakistan. We also studied the effect of gestational age, gender, weight and time of sampling on RIs of these markers extracted from DBS. Under the circumstances in Pakistan, where groups are working to evolve NBS, establishing RI on DBS from neonates from birth to 4 days of age is timely and needed for describing clinical association with specific diseases and effective clinical interventions.

## Materials and methods

### Study design

This cross-sectional community based descriptive study was conducted on a cohort of neonates enrolled in the AMANHI (Alliance for Maternal and Newborn Health Improvement) bio-repository at the AKU, Pakistan [19]. Briefly, the AMANHI biorepository study is a population-based pregnancy newborn cohort study, in three sites in South Asia and Africa [19]. The objective was to study the interactions between genes and biomarkers and a wide range of environmental exposures in causing diseases in pregnant women and newborns in low- and middle-income settings using uniform protocols and trained physicians [19]. The study was conducted in accordance with the Declaration of Helsinki Ethical Principles and Good Clinical Practices. The study received ethical approval from the Aga Khan University's Ethics Review Committee (Approval Number: 0591–3417) and World Health Organization's Ethics Review Committee (Approval number: RPC532). Written informed consent was taken before sample

collection from parents/ guardians of the newborns and ethical review committee approval was sought.

## Study population & data collection

Sampling was done from Nov 2017- Feb 2019 in peri-urban communities of Ibrahim Hyderi and Ali Akbar Shah. Using harmonized protocols, approximately two thousand five hundred women in their early pregnancies (8–19 weeks) were enrolled and followed throughout gestation, until 42 days' post-partum. All pregnancies were confirmed using urine pregnancy test followed by confirmation with sonography to accurately determine the gestation age. Detailed phenotypic and epidemiological data from pregnant women and their families were collected by trained field workers at scheduled household visits during pregnancy (enrolment, 24–28 weeks, 32–36 weeks & 38+ weeks) and post-partum (0–6 and 42–60 days post-delivery) [20].

DBS samples were obtained from heel prick (from the middle part of the heel) of babies of apparently healthy mothers within 6 days of birth. Blood was collected on Whatman 903 Protein Saver filter paper by trained phlebotomists, dried and stored in a biorepository following routine clinical care according to the Clinical Laboratory Standards Institute (CLSI C28-A3) guidelines [21]. A specially designed software was used for capturing data on biological samples which also guided the chronological flow of the sample collection process [20,22]. Babies born to mothers with metabolic diseases like gestational diabetes, preeclampsia, infections, and neonates who had any congenital anomaly were excluded from the study to avoid confounding effects. Neonates who had history of hospital admissions or did not survive on 60 days follow up were excluded from the study. Those neonates with incomplete datasets were also excluded.

## Mass spectrometer analysis

Samples were coded and transported to University of Iowa, Iowa city Indianapolis United States of America for analysis [23]. Analysis was performed on Quattro Micro triple quadruple tandem mass spectrometers from Waters, Eschborn, Germany. An electrospray ionization source was used using previously established methodology [24,25]. Briefly, butyl esters of analytes were prepared from the extracts by derivatization. Multiple reaction monitoring (MRM) mode was used to scan for specific mass ion intensities. Specific analytes concentrations were attained from the ratio of ion intensity at the mass and compared to its isotopically labeled internal standard and correcting for blood volume in a 1/8-inch DBS punch. Both internal and external spiked control specimens (from the Newborn Screening Quality Assurance Program at the Centers for Disease Control), a normal control specimen from healthy children, and a blank was analyzed with each batch of specimens. The intra-assay coefficient of variation (CV) for each biochemical analyte was less than 20%.

## Statistical analysis and determination of reference interval

The NCSS, SPSS version 21 and Stata v-14 were used for statistical analysis. Tests for normality of analyte distributions (Kolmogorov–Smirnov, Shapiro-Wilk) were carried out using SPSS 21 (IBM). Descriptive statistics were used for characteristics of study participants and concentrations of amino acids, succinylacetone and acyl carnitines. Mean values and standard deviations (SD) were calculated for normally distributed variables. Percentages were used to describe dichotomous variables. Chi-squared tests were used for dichotomous variables with statistical significance $\alpha$ set at 5%. Reference intervals were determined non-parametrically as most of the analytes were not normally distributed. The CLSI recommended method was used for the determination of upper and lower end points covering 95% of the reference values of each

analyte with respective 90% Confidence intervals (CI) [14]. Comparison of analyte distributions across categorical variables such as gender and GA were performed using the two sample independent samples t-test.

As per World Health Organization (WHO) neonates with weight less than 2500 gm (up to and including 2499 gm) were categorized as low birth weight (LBW) [26]. Babies were categorized into appropriate for gestational age (AGA) when weight was between 10th and 90th percentile in relation to gestational age, and into small for gestational age (SGA) when a baby's weight was less than the 10th percentile according to gender specific WHO growth chart for Asians [26]. Using WHO classification neonates were also classified according to gestational age (GA) as early term or premature, full term, and post term; if born before 37 weeks, between 38 to 42 weeks and beyond 42 weeks of pregnancy respectively. To further assess the effect of gender on the RI of markers the data was divided into 3 groups and RI compared also taking GA and neonatal weight into account. Group I included AGA neonates. Preterm and post term neonates were not excluded from this group. Group II included full term and SGA neonates. All preterm and post term neonates were excluded from this group. Group III had AGA and full-term neonates only. All preterm, post term and SGA neonates were excluded from Group III. As a measure of association, we used the F test comparing differences across AGA/SGA, and preterm, term and post term newborns under the null hypothesis that all groups had the same mean response. RIs were calculated before and after adding SGA newborns and preterm babies (<37 weeks).

The authors agreed that the statistical significance indicates the reliability of the study results while the clinical significance reflects its impact on clinical practice. Based on this the amino acids and acylcarnitine with statistically significant differences were individually analyzed by three subject experts. The subject experts based on their experiences and literature review noted their decision. This was followed by developing consensus in a meeting regarding the bias (p value) being clinically significant (even if any one expert disagreed) or insignificant (if all agreed).

## Results

A total of 635 DBS were analyzed in the laboratory at the University of Iowa. Twenty-five DBS were excluded based on the defined exclusion criteria, leaving a total of 610 newborn DBS available for analyses.

## Demographics and clinical characteristics of maternal and neonatal cohort

Out of the total 610 DBS, 325 (53.2%) were from females. Samples were representative of all ethnicities of Pakistan including Urdu speaking (37.9%), Sindhi (24.9%), Bengali (19.7%), Punjabi (9.7%), Pathan (2.1%), Baloch (1.6%) and other minorities (4.1%). Mean GA of mothers at the time of delivery was 38.4±1.5 weeks. Mean birth weight was 2789.9±468.3 grams, 145 (23.7%) neonates being LBW, 219 (35.9%) were SGA and 87 (14.3%) being preterm. A summary of demographics, mothers' GA, maturity, and birth weight of neonates is provided in Table 1.

## Distribution of amino acids, acyl carnitines and succinylacetone concentrations and their RIs in neonates

The mean and median levels and parametric RIs of eleven amino acids and their ratios, succinylacetone and thirty-two acylcarnitines (short, medium, and long chain) and their twelve ratios in DBS neonatal blood samples are summarized in Table 2. The concentrations of

**Table 1. Demographics and clinical characteristics of maternal and neonatal cohort included in the AMANHI biorepository.**

| Variables | | Overall n = 610 | Male n = 285 | Female n = 325 | p value |
|---|---|---|---|---|---|
| **Maternal cohort demographics and clinical history** | | | | | |
| Mean age (years) | | 26.8 ±0.2 | 26.7 ±5.5 | 26.8 ±5.4 | 0.8792 |
| Mean BMI (kg/m2) | | 22.5 ±0.2 | 22.6 ±4.9 | 22.3 ±4.7 | 0.5035 |
| Mean BMI category according to South Asian Classification, n (%) | <18.5 kg/m2 | 134 (22) | 54 (19) | 80 (24.6) | 0.284 |
| | <18.5–22.9 kg/m2 | 232 (38) | 113 (39.7) | 119 (36.6) | |
| | 23–26.9 kg/m2 | 136 (22.3) | 62 (21.8) | 74 (22.8) | |
| | >27 kg/m2 | 108 (17.7) | 56 (19.7) | 52 (16) | |
| Low (<23cm) mid-upper arm circumference (MUAC), n (%) | | 248 (40.7) | 108 (38) | 140 (43.1) | 0.206 |
| Prior history of still births, n (%) | | 29 (4.8) | 14 (4.9) | 15 (4.6) | 0.985 |
| History of diabetes, n (%) | | 5 (0.8) | 3 (1.1) | 2 (0.6) | 0.669 |
| **Neonatal cohort demographics and clinical history** | | | | | |
| Mean gestational age (weeks) | | 38.7±1.6 | 38.7 ±1.7 | 38.8 ±1.5 | 0.1408 |
| Gestational age category, n (%) | Preterm (<37 Weeks) | 87 (14.3) | 46 (16.1) | 41 (12.6) | 0.292 |
| | Term (37–42 Weeks) | 515 (84.4) | 234 (82.1) | 281 (86.5) | |
| | Post Term (>42 Weeks) | 8 (1.3) | 5 (1.8) | 3 (0.9) | |
| Birth weight in gm, mean±SD | | 2778.8±472.9 | 2833.1 ±521.4 | 2731.5 ±421.4 | 0.0081 |
| Categorization of neonate's weight, n (%) | Normal Weight | 465 (76.2) | 219 (77.1) | 246 (75.7) | 0.723 |
| | Low Birth Weight | 145 (23.7) | 66 (22.9) | 79 (24.3) | 0.656 |
| Categorization of neonate's weight in relation to gestational age, n (%) | Appropriate for gestational age | 391 (64.0) | 181 (46.3) | 210 (53.7) | <0.0001 |
| | Small for gestational age | 219 (35.9) | 104 (47.5) | 115 (52.5) | <0.0001 |

Demographics and clinical data were presented as numbers (percentages). Chi-square analysis was carried out to determine the frequency distribution amongst groups. For quantitative data mean comparison was done using t-test and ANOVA between two groups and more than two groups respectively. A p-value less than 0.05 was taken as statistically significant.

argininosuccinic acid and succinylacetone were <1 μmol/L (micromole per liter) in the blood spot extracts.

## Effect of gestational age and birth weight on gender based RIs

The influence of gender on the amino acids and succinylacetone studies was examined and no statistically significant difference (p value >0.05) was found as described in Fig 1A and Table 2. Short chain acylcarnitine that showed statistically significant difference when examined by gender were C0, C2, C3-DC, C4-DC, C4-OH, C5:1 and C5-DC. However, this difference was clinically insignificant as the 97.5th percentile was similar in males and females as shown in Table 2. As described for C5-DC, the 97.5[th] percentile for C5-DC was almost similar in males (0.08 μmol/L) and females (0.07 μmol/L). All short chain acylcarnitne (C6-C12) and many long chain acylcarnitines (C14, C14:1, C14:2, C14-OH, C16 and C18:2) showed statistically significant difference when examined by gender; being higher in males again with clinically insignificant difference (Fig 1B and Table 2). Although statistically significant, the magnitude of the gender differences on acylcarnitines seemed not sufficiently large to specify the use of separate RI for clinical use.

S1 Table compares the effect of gender based on GA and gestational weight on the RIs of amino acids, succinylacetone and acylcarnitine concentrations after dividing into 3 groups.

**Table 2. Distribution of dried blood spot amino acids, acylcarnitines and succinylacetone concentrations in neonates included in AMANHI study.**

| Analytes/ Markers with cutoffs (µmol/L) | | Overall, n = 610 | | | Gender Difference | | | | | | p-value |
|---|---|---|---|---|---|---|---|---|---|---|---|
| | | | | | Male, n = 285 | | | Female, n = 325 | | | |
| | | Mean | Median | 2.5%–97.5% | Mean | Median | 2.5%–97.5% | Mean | Median | 2.5%–97.5% | |
| **Amino Acids** | **Alanine** | 237.26 | 219.53 | 121.05–461.52 | 236.02 | 218.44 | 120.69–482.25 | 238.35 | 219.95 | 120.63–436.48 | 0.7432 |
| | **Arginine** | 4.3 | 3.92 | 1.58–9.02 | 4.39 | 3.99 | 1.57–10.4 | 4.23 | 3.88 | 1.61–8.91 | 0.3525 |
| | **Arginosuccininc acid** | 0.02 | 0.02 | 0.01–0.03 | 0.02 | 0.02 | 0.01–0.04 | 0.02 | 0.02 | 0.01–0.03 | 0.7820 |
| | **Citrulline** | 10.94 | 10.47 | 5.94–18.87 | 10.77 | 10.25 | 5.76–19.92 | 11.08 | 10.75 | 6–18.75 | 0.2807 |
| | **Glutamine** | 285.85 | 285.85 | 167.25–461.51 | 291.74 | 282.18 | 159.95–463.31 | 299.29 | 289.14 | 172.73–456.77 | 0.2293 |
| | **Leucine** | 176.26 | 169.04 | 92–295.22 | 179.05 | 170.72 | 90.71–301.06 | 173.8 | 167.45 | 91.94–291.05 | 0.218 |
| | **Methionine** | 20.44 | 19.41 | 11.33–35.48 | 20.77 | 19.26 | 11.68–40.68 | 20.14 | 19.43 | 11.2–33.58 | 0.2185 |
| | **Ornithine** | 31.71 | 30.38 | 14.02–53.31 | 31.82 | 30.22 | 13.41–54.97 | 31.62 | 30.71 | 14–53.32 | 0.8211 |
| | **Phenylalanine** | 78.06 | 75.27 | 42.45–123.87 | 77.84 | 74.99 | 42.12–122.65 | 78.25 | 74.87 | 43.12–124.71 | 0.719 |
| | **Tyrosine** | 99.79 | 90.05 | 47.77–213.38 | 102.95 | 90.85 | 46.9–243.79 | 97.01 | 88.81 | 48.04–207.26 | 0.1308 |
| | **Valine** | 117.08 | 115.5 | 65.51–188.87 | 116.20 | 114.03 | 64.46–191.24 | 117.86 | 116.4 | 66.32–185.37 | 0.5102 |
| **Amino Acid Ratios** | **Arginine/Ornithine** | 0.14 | 0.13 | 0.05–0.32 | 0.14 | 0.14 | 0.05–0.35 | 0.14 | 0.13 | 0.06–0.31 | 0.8690 |
| | **Citrulline/Arginine** | 3.01 | 2.7 | 1.01–6.94 | 2.93 | 2.68 | 0.94–6.62 | 3.08 | 2.76 | 1.12–7.22 | 0.235 |
| | **Leucine/Alanine** | 0.79 | 0.77 | 0.41–1.41 | 0.81 | 0.79 | 0.41–1.46 | 0.77 | 0.75 | 0.41–1.25 | 0.045 |
| | **Leucine/Phenylalanine** | 2.37 | 2.25 | 1.5–3.87 | 2.40 | 2.29 | 1.44–3.91 | 2.33 | 2.21 | 1.54–3.89 | 0.191 |
| | **Methionine/ Phenylalanine** | 0.27 | 0.26 | 0.18–0.41 | 0.28 | 0.26 | 0.19–0.42 | 0.27 | 0.26 | 0.18–0.4 | 0.041 |
| | **Phenylalanine/Tyrosine** | 0.88 | 0.84 | 0.32–1.6 | 0.86 | 0.81 | 0.33–1.52 | 0.91 | 0.86 | 0.31–1.72 | 0.078 |
| | **Tyrosine/Phenylalanine** | 1.33 | 1.2 | 0.63–3.16 | 1.36 | 1.25 | 0.66–3.03 | 1.30 | 1.16 | 0.59–3.24 | 0.29 |
| **Organic acid** | **Succinylacetone** | 0.73 | 0.72 | 0.5–1.02 | 0.73 | 0.72 | 0.49–1.05 | 0.72 | 0.71 | 0.5–1 | 0.5135 |
| **Short chain acylcarnitines** | **C0** | 26.85 | 25.64 | 12.45–49.46 | 28.30 | 26.95 | 13.14–49.52 | 25.57 | 24.01 | 12.22–49.5 | 0.0003 |
| | **C2** | 24.44 | 23.56 | 9.77–45.15 | 25.68 | 24.0 | 10.27–47.08 | 23.35 | 22.5 | 9.45–41.48 | 0.0014 |
| | **C3** | 2.13 | 1.9 | 0.74–4.78 | 2.18 | 2.0 | 0.77–4.86 | 2.10 | 1.8 | 0.73–4.78 | 0.3654 |
| | **C3-DC** | 0.03 | 0.03 | 0.02–0.06 | 0.03 | 0.03 | 0.02–0.06 | 0.03 | 0.03 | 0.02–0.05 | <0.0001 |
| | **C4** | 0.28 | 0.25 | 0.1–0.67 | 0.28 | 0.27 | 0.1–0.69 | 0.27 | 0.25 | 0.1–0.68 | 0.3385 |
| | **C4-DC** | 0.15 | 0.14 | 0.06–0.28 | 0.15 | 0.14 | 0.06–0.29 | 0.14 | 0.14 | 0.06–0.28 | 0.0404 |
| | **C4-OH** | 0.14 | 0.13 | 0.05–0.32 | 0.15 | 0.14 | 0.06–0.33 | 0.14 | 0.13 | 0.05–0.3 | 0.0010 |
| | **C5** | 0.14 | 0.12 | 0.05–0.31 | 0.13 | 0.14 | 0.06–0.31 | 0.14 | 0.13 | 0.05–0.34 | 0.3363 |
| | **C5:1** | 0.01 | 0.01 | 0.01–0.02 | 0.01 | 0.01 | 0.01–0.02 | | 0.01 | 0.01–0.02 | 0.0415 |
| | **C5-DC** | 0.04 | 0.03 | 0.01–0.08 | 0.04 | 0.03 | 0.02–0.08 | 0.03 | 0.03 | 0.01–0.07 | <0.0001 |
| | **C5-OH** | 0.1 | 0.09 | 0.05–0.17 | 0.1 | 0.09 | 0.06–0.17 | 0.1 | 0.09 | 0.05–0.19 | 0.0871 |
| **Medium chain acylcarnitines** | **C6** | 0.06 | 0.05 | 0.02–0.11 | 0.06 | 0.05 | 0.02–0.12 | 0.05 | 0.04 | 0.02–0.11 | 0.0002 |
| | **C6-DC** | 0.02 | 0.01 | 0.01–0.03 | 0.02 | 0.02 | 0.01–0.03 | 0.01 | 0.01 | 0.01–0.03 | 0.0037 |
| | **C8** | 0.07 | 0.06 | 0.03–0.16 | 0.08 | 0.07 | 0.03–0.17 | 0.06 | 0.05 | 0.03–0.15 | <0.0001 |
| | **C8:1** | 0.1 | 0.08 | 0.02–0.25 | 0.11 | 0.09 | 0.02–0.26 | 0.09 | 0.07 | 0.02–0.24 | 0.0037 |
| | **C10** | 0.1 | 0.08 | 0.04–0.25 | 0.11 | 0.09 | 0.04–0.26 | 0.09 | 0.08 | 0.03–0.2 | 0.0002 |
| | **C10:1** | 0.06 | 0.05 | 0.02–0.16 | 0.07 | 0.06 | 0.03–0.18 | 0.06 | 0.05 | 0.02–0.13 | <0.0001 |
| | **C12** | 0.2 | 0.18 | 0.08–0.44 | 0.22 | 0.20 | 0.09–0.44 | 0.19 | 0.18 | 0.07–0.41 | 0.0006 |
| | **C12:1** | 0.11 | 0.09 | 0.03–0.3 | 0.12 | 0.09 | 0.03–0.36 | 0.10 | 0.08 | 0.03–0.27 | 0.0025 |

(*Continued*)

**Table 2.** (Continued)

| Analytes/ Markers with cutoffs (µmol/L) | | Overall, n = 610 | | | Gender Difference | | | | | | *p*-value |
|---|---|---|---|---|---|---|---|---|---|---|---|
| | | | | | Male, n = 285 | | | Female, n = 325 | | | |
| | | Mean | Median | 2.5%–97.5% | Mean | Median | 2.5%–97.5% | Mean | Median | 2.5%–97.5% | |
| Long chain acylcarnitines | C14 | 0.32 | 0.3 | 0.16–0.56 | 0.35 | 0.33 | 0.17–0.57 | 0.30 | 0.3 | 0.15–0.55 | <0.0001 |
| | C14:1 | 0.16 | 0.15 | 0.06–0.34 | 0.18 | 0.16 | 0.08–0.35 | 0.15 | 0.14 | 0.06–0.32 | <0.0001 |
| | C14:2 | 0.02 | 0.02 | 0.01–0.04 | 0.02 | 0.02 | 0.01–0.04 | 0.02 | 0.02 | 0.01–0.04 | <0.0001 |
| | C14-OH | 0.02 | 0.02 | 0.01–0.04 | 0.02 | 0.16 | 0.01–0.04 | 0.02 | 0.14 | 0.01–0.04 | 0.0001 |
| | C16 | 3.96 | 3.8 | 1.97–6.89 | 4.11 | 3.9 | 2.04–7.22 | 3.82 | 3.7 | 1.92–6.69 | 0.0050 |
| | C16-OH | 0.02 | 0.02 | 0.01–0.05 | 0.03 | 0.02 | 0.01–0.05 | 0.02 | 0.02 | 0.01–0.05 | 0.0001 |
| | C16:1 | 0.25 | 0.24 | 0.11–0.44 | 0.26 | 1.10 | 0.12–0.43 | 0.24 | 1.01 | 0.09–0.45 | 0.0013 |
| | C16:1-OH | 0.05 | 0.04 | 0.02–0.09 | 0.05 | 0.05 | 0.02–0.1 | 0.04 | 0.04 | 0.02–0.08 | <0.0001 |
| | C18 | 1.11 | 1.04 | 0.58–2.06 | 1.15 | 1.10 | 0.53–2.13 | 1.07 | 1.01 | 0.6–1.98 | 0.0170 |
| | C18-OH | 0.01 | 0.01 | 0.01–0.03 | 0.01 | 0.02 | 0.01–0.02 | 0.01 | 0.02 | 0.01–0.03 | 0.0020 |
| | C18:1 | 1.5 | 1.44 | 0.76–2.58 | 1.57 | 0.03 | 0.74–2.78 | 1.43 | 0.03 | 0.77–2.44 | 0.0001 |
| | C18:2 | 0.16 | 0.14 | 0.06–0.36 | 0.17 | 0.16 | 0.07–0.39 | 0.15 | 0.13 | 0.06–0.35 | 0.0004 |
| | C18:1-OH | 0.02 | 0.02 | 0.01–0.03 | 0.02 | 0.02 | 0.01–0.03 | 0.02 | 0.02 | 0.01–0.03 | 0.0001 |
| Acylcarnitine Ratios | C0/C16 | 7.1 | 6.51 | 3.62–13.87 | 7.23 | 6.9 | 3.71–14.36 | 6.99 | 6.6 | 3.57–13.8 | 0.2292 |
| | C0/C18 | 24.89 | 23.13 | 12.13–47.16 | 25.57 | 23.31 | 12.91–47.85 | 24.29 | 23.03 | 12.04–46.95 | 0.0915 |
| | C3/C2 | 0.09 | 0.08 | 0.04–0.18 | 0.09 | 0.08 | 0.04–0.16 | 0.09 | 0.09 | 0.04–0.19 | 0.5561 |
| | C4/C2 | 0.01 | 0.01 | 0.01–0.03 | 0.01 | 0.01 | 0.01–0.03 | 0.01 | 0.01 | 0.01–0.03 | 0.4972 |
| | C4/C3 | 0.15 | 0.13 | 0.06–0.35 | 0.15 | 0.14 | 0.06–0.36 | 0.15 | 0.13 | 0.06–0.35 | 0.7361 |
| | C5/C2 | 0.01 | 0.01 | 0–0.01 | 0.00 | 0.01 | 0–0.01 | 0.01 | 0.01 | 0–0.02 | 0.0007 |
| | C5/C3 | 0.07 | 0.06 | 0.03–0.16 | 0.07 | 0.06 | 0.03–0.15 | 0.07 | 0.06 | 0.03–0.17 | 0.0459 |
| | C5-DC/C8 | 0.54 | 0.52 | 0.31–0.87 | 0.52 | 0.50 | 0.29–0.9 | 0.55 | 0.54 | 0.33–0.85 | 0.0147 |
| | C5-DC/C16 | 0.01 | 0.01 | 0–0.02 | 0.01 | 0.01 | 0–0.02 | 0.01 | 0.01 | 0–0.02 | 0.0591 |
| | C8/C10 | 0.73 | 0.72 | 0.52–0.99 | 0.74 | 0.73 | 0.53–1.02 | 0.72 | 0.71 | 0.52–0.99 | 0.1655 |
| | C14:1/C16 | 0.04 | 0.04 | 0.02–0.08 | 0.04 | 0.05 | 0.02–0.09 | 0.04 | 0.04 | 0.02–0.08 | 0.0026 |
| | C14:1/C12:1 | 1.75 | 1.67 | 0.99–3 | 1.76 | 1.69 | 0.96–3.11 | 1.74 | 1.65 | 1.02–2.91 | 0.5081 |
| | C16-OH/C16 | 0.01 | 0.01 | 0–0.01 | 0.01 | 0.01 | 0–0.01 | 0.01 | 0.01 | 0–0.01 | 0.3531 |

*p-values are calculated using Mann Whitney t-test amongst gender differences. A p-value less than 0.05 was taken as statistically significant.

Group I, II & III comprised of 391, 515 and 334 AGA neonates, respectively. The magnitude of the gender differences on markers studied was not sufficiently large to specify the clinical use of separate RI for both genders (ST1).

## Effect of gestational age and neonatal weight

Table 3 describes the relative difference in DBS amino acids, succinylacetone and acylcarnitine concentrations between preterm and full term groups (taking the full term group as a reference) and GA (taking the AGA group as a reference). Post term babies were excluded from the analysis as they were few (n = 8). Mean alanine (240.5±88.1 versus 218.4±83.6 µmol/L; p value 0.03), valine (118.6±31.3 versus 108.8±28.2 µmol/L; p value 0.007) and C4-DC (0.15±0.06 versus 0.12±0.04 µmol/L; p value<0.001) were slightly higher in full term neonates as compared to pre terms. Mean difference of 21.6 µmol/L, 9.8 µmol/L and 0.03 µmol/L was noted in alanine, valine and C4-DC respectively in full term as compared to pre term neonates. Mean difference, being higher in premature neonates, was noted in arginine (0.51 µmol/L), tyrosine (27.63 µmol/L), C3 (0.53 µmol/L), C3DC (0.01 µmol/L), C4 (0.51 µmol/L), C4OH (0.03 µmol/

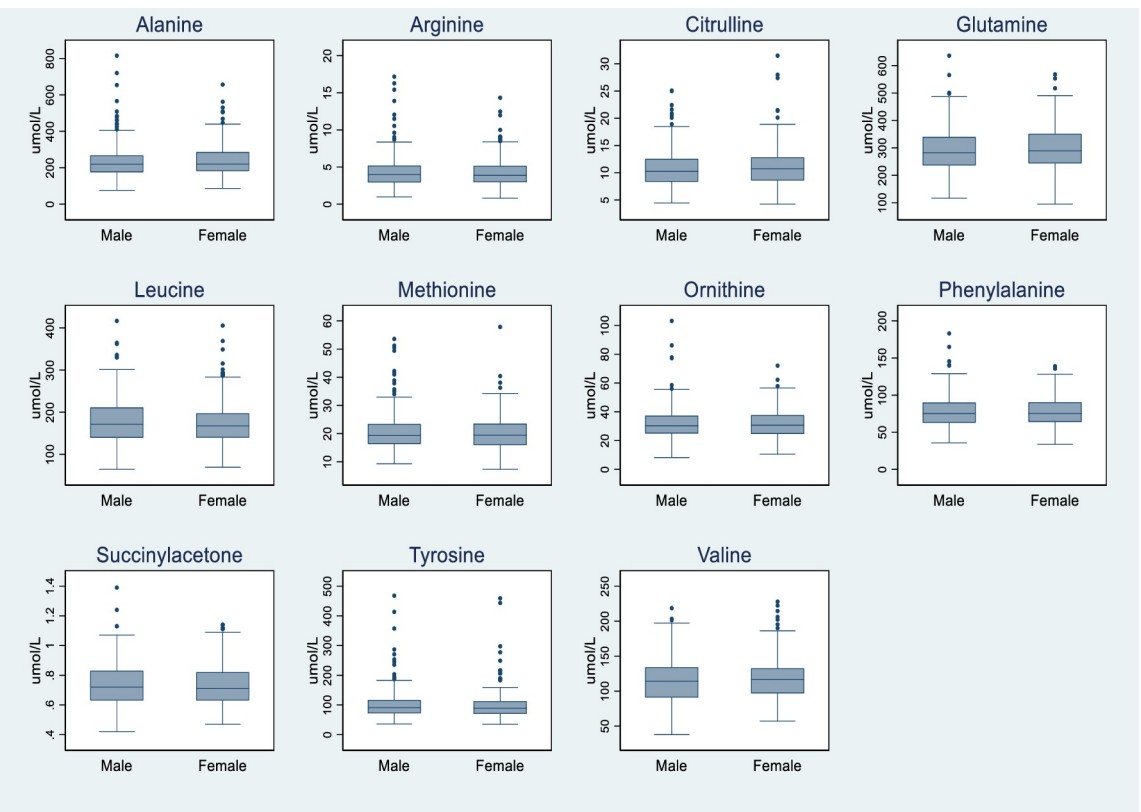

**Fig 1. Gender wise distribution of DBS biomarkers in cohort of neonates from AMANHI biorepository.** a: Gender wise distribution of DBS amino acid and succinylacetone concentrations in cohort of neonates from AMANHI biorepository in Pakistan (n = 610). b: Gender wise distribution of DBS acylcarnitine in cohort of neonates from AMANHI biorepository in Pakistan (n = 610).

L), C5 (0.04 μmol/L), C5DC (0.01 μmol/L), C6DC (0.01 μmol/L), C10 (0.03 μmol/L), C10:1 (0.01 μmol/L), C12:1 (0.04 μmol/L), C14 (0.07 μmol/L), C14:1 (0.05 μmol/L), C14:2 (0.01 μmol/L), C14OH (0.01 μmol/L), C16OH (0.01 μmol/L) and C18OH (0.01 μmol/L), ($p$ value $<0.05$).

Neonates who were SGA showed lower mean alanine (232.16±80.64 versus 247.7 ±100.3 μmol/L; $p$ value 0.04), arginine (4.1±1.9 versus 4.6±2.2 μmol/L; p value 0.006), citrulline (10.6±3.3 versus 11.3±3.8; $p$ value 0.023), ornithine (30.7±9.8 versus 33.5±11.5; $p$ value 0.002), C0 (26.6±9.0 versus 27.0±9.8; $p$ value 0.585) and C10:1 (0.06±0.03 versus 0.07±0.04; $p$ value 0.000) as compared to neonates who were AGA. The mean difference in these amino acids and acylcarnitine is clinically minor except for alanine which is higher in AGA in comparison to SGA neonates with a mean difference of ±15 umol/L and higher in full terms as compared to preterm with a mean difference of ±22 umol/L.

## Effect of time of sampling

Graphic forms showing trends of the amino acids, acylcarnitines and succinylacetone median concentrations according to time of collection are illustrated in Fig 2A and 2B. The concentrations of alanine (r = -0.04), citrulline (r = -0.22), methionine (r = -0.17), phenylalanine (r = -0.10) and tyrosine (r = -0.04) negatively correlated with age over the first 4 days of life but the correlation was weak (Fig 2).

**Table 3. Relative difference in amino acids, succinylacetone and acylcarnitine concentrations between term groups and gestational age in dried blood spot of neonates from AMANHI study.**

| Analytes/ Markers | | Mean ± SD in μmol/L Median (2.5% - 97.5%) Overall, n = 610 | Mean difference in concentration in μmol/L | | | Mean difference in concentration in μmol/L | | |
|---|---|---|---|---|---|---|---|---|
| | | | Preterm | Full Term | p value | SGA | AGA | p value |
| | | | n = 87 | n = 515 | | n = 219 | n = 391 | |
| Amino Acids | Alanine | 237.4 ± 87.8 219.58 (120.37–461.96) | 218.49 ± 83.66 | 240.57 ± 88.12 | 0.030 | 232.16 ± 80.64 | 247.71 ± 100.32 | 0.041 |
| | Arginine | 4.3 ± 2.1 3.91 (1.58–9.03) | 4.74 ± 2.72 | 4.23 ± 1.95 | 0.034 | 4.17 ± 1.96 | 4.65 ± 2.26 | 0.006 |
| | Arginosuccinic acid | 0.02 ± 0.007 0.02 (0.01–0.03) | 0.02 ± 0.01 | 0.02 ± 0.01 | 0.582 | 0.02 ± 0.01 | 0.02 ± 0.01 | 0.069 |
| | Citrulline | 10.9 ± 3.5 10.47 (5.92–18.89) | 10.38 ± 3.07 | 11.04 ± 3.59 | 0.107 | 10.68 ± 3.32 | 11.37 ± 3.89 | 0.023 |
| | Glutamine | 296 ± 77.5 285.85 (167.1–461.96) | 289.34 ± 87.36 | 297.17 ± 75.7 | 0.383 | 296.35 ± 78.84 | 293.2 ± 74.16 | 0.637 |
| | Leucine | 176.4 ± 52.4 169.49 (91.81–295.45) | 182.2 ± 60.99 | 175.41 ± 50.83 | 0.265 | 176.53 ± 52.08 | 176.91 ± 54.27 | 0.934 |
| | Methionine | 20.4 ± 6.4 19.41 (11.32–35.58) | 21.25 ± 7.55 | 20.31 ± 6.12 | 0.201 | 20.35 ± 6.03 | 20.62 ± 6.96 | 0.629 |
| | Ornithine | 31.7 ± 10.6 30.46 (13.99–53.32) | 29.94 ± 10.52 | 32.05 ± 10.54 | 0.084 | 30.75 ± 9.86 | 33.58 ± 11.51 | 0.002 |
| | Phenylalanine | 78.1 ± 21.1 75.44 (42.32–124.11) | 76.05 ± 18.45 | 78.48 ± 21.52 | 0.322 | 77.03 ± 20.77 | 79.94 ± 21.65 | 0.111 |
| | Tyrosine | 100 ± 48.6 90.05 (47.83–214.99) | 123.69 ± 67.05 | 96.06 ± 43.56 | <0.001 | 98.28 ± 45.62 | 103.32 ± 54.43 | 0.232 |
| | Valine | 117.2 ± 31.1 115.81 (65.4–189.06) | 108.84 ± 28.2 | 118.61 ± 31.33 | 0.007 | 116.04 ± 29.98 | 118.31 ± 32.87 | 0.399 |
| Organic acid | Succinylacetone | 0.727 ± 0.138 0.71 (0.5–1.03) | 0.74 ± 0.16 | 0.72 ± 0.13 | 0.232 | 0.72 ± 0.14 | 0.74 ± 0.14 | 0.058 |
| Acylcarnitine | C0 | 26.8 ± 9.3 25.63 (12.45–49.12) | 27.5 ± 9.67 | 26.65 ± 9.26 | 0.429 | 26.62 ± 9.09 | 27.06 ± 9.84 | 0.585 |
| | C2 | 24.4 ± 9.01 23.52 (9.73–44.99) | 25.93 ± 9.24 | 24.1 ± 8.95 | 0.081 | 23.97 ± 9.1 | 25.48 ± 8.9 | 0.053 |
| | C3 | 2.1 ± 1.072 1.9 (0.73–4.79) | 2.59 ± 1.41 | 2.06 ± 0.99 | <0.001 | 2.19 ± 1.06 | 2.08 ± 1.11 | 0.222 |
| | C3-DC | 0.032 ± 0.011 0.03 (0.02–0.06) | 0.04 ± 0.01 | 0.03 ± 0.01 | <0.001 | 0.03 ± 0.01 | 0.03 ± 0.01 | 0.004 |
| | C4 | 0.275 ± 0.143 0.24 (0.1–0.68) | 0.32 ± 0.13 | 0.27 ± 0.14 | 0.002 | 0.27 ± 0.14 | 0.28 ± 0.16 | 0.403 |
| | C4-DC | 0.147 ± 0.055 0.14 (0.06–0.27) | 0.12 ± 0.04 | 0.15 ± 0.06 | <0.001 | 0.14 ± 0.05 | 0.15 ± 0.06 | 0.133 |
| | C4-OH | 0.143 ± 0.064 0.13 (0.05–0.32) | 0.17 ± 0.07 | 0.14 ± 0.06 | <0.001 | 0.14 ± 0.06 | 0.15 ± 0.07 | 0.037 |
| | C5 | 0.138 ± 0.072 0.12 (0.05–0.31) | 0.17 ± 0.09 | 0.13 ± 0.07 | <0.001 | 0.14 ± 0.07 | 0.14 ± 0.07 | 0.610 |
| | C5:1 | 0.012 ± 0.004 0.01 (0.01–0.02) | 0.01 ± 0.004 | 0.01 ± 0.004 | 0.001 | 0.01 ± 0.004 | 0.01 ± 0.004 | 0.953 |
| | C5-DC | 0.035 ± 0.016 0.03 (0.01–0.08) | 0.04 ± 0.02 | 0.03 ± 0.01 | <0.001 | 0.03 ± 0.02 | 0.04 ± 0.02 | 0.008 |
| | C5-OH | 0.098 ± 0.032 0.09 (0.05–0.17) | 0.1 ± 0.03 | 0.1 ± 0.03 | 0.120 | 0.1 ± 0.03 | 0.1 ± 0.03 | 0.098 |

(*Continued*)

**Table 3.** (Continued)

| Analytes/ Markers | | Mean ± SD in μmol/L Median (2.5% - 97.5%) Overall, n = 610 | Mean difference in concentration in μmol/L | | | Mean difference in concentration in μmol/L | | |
|---|---|---|---|---|---|---|---|---|
| | | | Preterm | Full Term | *p* value | SGA | AGA | *p* value |
| | | | n = 87 | n = 515 | | n = 219 | n = 391 | |
| | C6 | 0.056 ± 0.025 0.05 (0.02–0.11) | 0.06 ± 0.03 | 0.05 ± 0.02 | 0.031 | 0.05 ± 0.03 | 0.06 ± 0.03 | 0.026 |
| | C6-DC | 0.015 ± 0.007 0.01 (0.01–0.03) | 0.02 ± 0.01 | 0.01 ± 0.01 | 0.001 | 0.02 ± 0.01 | 0.02 ± 0.01 | 0.784 |
| | C8 | 0.068 ± 0.036 0.06 (0.03–0.15) | 0.08 ± 0.04 | 0.07 ± 0.03 | <0.001 | 0.07 ± 0.04 | 0.07 ± 0.05 | 0.023 |
| | C8:1 | 0.097 ± 0.065 0.08 (0.02–0.25) | 0.1 ± 0.08 | 0.1 ± 0.06 | 0.429 | 0.09 ± 0.06 | 0.11 ± 0.08 | 0.003 |
| | C10 | 0.096 ± 0.058 0.08 (0.04–0.24) | 0.12 ± 0.08 | 0.09 ± 0.05 | 0.000 | 0.09 ± 0.06 | 0.11 ± 0.07 | 0.013 |
| | C10:1 | 0.063 ± 0.034 0.05 (0.02–0.15) | 0.07 ± 0.04 | 0.06 ± 0.03 | 0.031 | 0.06 ± 0.03 | 0.07 ± 0.04 | 0.000 |
| | C12 | 0.202 ± 0.099 0.18 (0.08–0.43) | 0.24 ± 0.13 | 0.2 ± 0.09 | <0.001 | 0.2 ± 0.1 | 0.21 ± 0.09 | 0.189 |
| | C12:1 | 0.106 ± 0.072 0.09 (0.03–0.29) | 0.14 ± 0.1 | 0.1 ± 0.07 | <0.001 | 0.1 ± 0.07 | 0.12 ± 0.08 | 0.022 |
| | C14 | 0.323 ± 0.121 0.3 (0.16–0.56) | 0.38 ± 0.15 | 0.31 ± 0.11 | <0.001 | 0.32 ± 0.13 | 0.33 ± 0.11 | 0.604 |
| | C14:1 | 0.164 ± 0.079 0.15 (0.06–0.34) | 0.21 ± 0.11 | 0.16 ± 0.07 | <0.001 | 0.16 ± 0.08 | 0.17 ± 0.08 | 0.188 |
| | C14:2 | 0.021 ± 0.01 0.02 (0.01–0.04) | 0.03 ± 0.02 | 0.02 ± 0.01 | <0.001 | 0.02 ± 0.01 | 0.02 ± 0.01 | 0.001 |
| | C14-OH | 0.022 ± 0.009 0.02 (0.01–0.04) | 0.03 ± 0.01 | 0.02 ± 0.01 | <0.001 | 0.02 ± 0.01 | 0.02 ± 0.01 | 0.169 |
| | C16 | 3.954 ± 1.288 3.8 (1.95–6.92) | 4.06 ± 1.13 | 3.94 ± 1.31 | 0.421 | 3.96 ± 1.3 | 3.92 ± 1.27 | 0.719 |
| | C16-OH | 0.025 ± 0.01 0.02 (0.01–0.05) | 0.03 ± 0.01 | 0.02 ± 0.01 | <0.001 | 0.02 ± 0.01 | 0.03 ± 0.01 | 0.209 |
| | C16:1 | 0.245 ± 0.083 0.24 (0.11–0.44) | 0.28 ± 0.08 | 0.24 ± 0.08 | <0.001 | 0.24 ± 0.08 | 0.25 ± 0.08 | 0.336 |
| | C16:1-OH | 0.047 ± 0.018 0.04 (0.02–0.09) | 0.05 ± 0.02 | 0.05 ± 0.02 | 0.755 | 0.05 ± 0.02 | 0.05 ± 0.02 | 0.765 |
| | C18 | 1.106 ± 0.386 1.04 (0.58–2.07) | 1.14 ± 0.36 | 1.1 ± 0.39 | 0.367 | 1.12 ± 0.39 | 1.08 ± 0.37 | 0.279 |
| | C18-OH | 0.013 ± 0.006 0.01 (0.01–0.03) | 0.02 ± 0.01 | 0.01 ± 0.01 | 0.003 | 0.01 ± 0.01 | 0.01 ± 0.01 | 0.153 |
| | C18:1 | 1.498 ± 0.459 1.44 (0.76–2.59) | 1.55 ± 0.45 | 1.49 ± 0.46 | 0.248 | 1.5 ± 0.46 | 1.49 ± 0.46 | 0.867 |
| | C18:2 | 0.157 ± 0.078 0.14 (0.06–0.36) | 0.16 ± 0.07 | 0.16 ± 0.08 | 0.622 | 0.15 ± 0.08 | 0.16 ± 0.08 | 0.265 |
| | C18:1-OH | 0.019 ± 0.006 0.02 (0.01–0.03) | 0.02 ± 0.01 | 0.02 ± 0.01 | 0.023 | 0.02 ± 0.01 | 0.02 ± 0.01 | 0.770 |

The full-term group is used as a reference. Differences from comparison across birth weight groups were determined using two sample t test. Mean ± SD (all such values) is given for overall and groups. Median (2.5–97.5th percentile) are given for overall data. p value of <0.05 was taken as statistically significant. Using WHO classification neonates were also classified based on GA as early term or premature: Born before 37 weeks of pregnancy are completed, full term: Born between 38 to 42 weeks and as post term: Born beyond 42 weeks of pregnancy. Please note that we have excluded the post term babies from the analysis as they were few (n = 8). Babies were categorized into appropriate for GA (AGA) when weight was between 10th and 90th percentile in relation to gestational age, and into small for GA (SGA) when a baby's weight was less than the 10th percentile according to gender specific WHO growth chart for Asians.

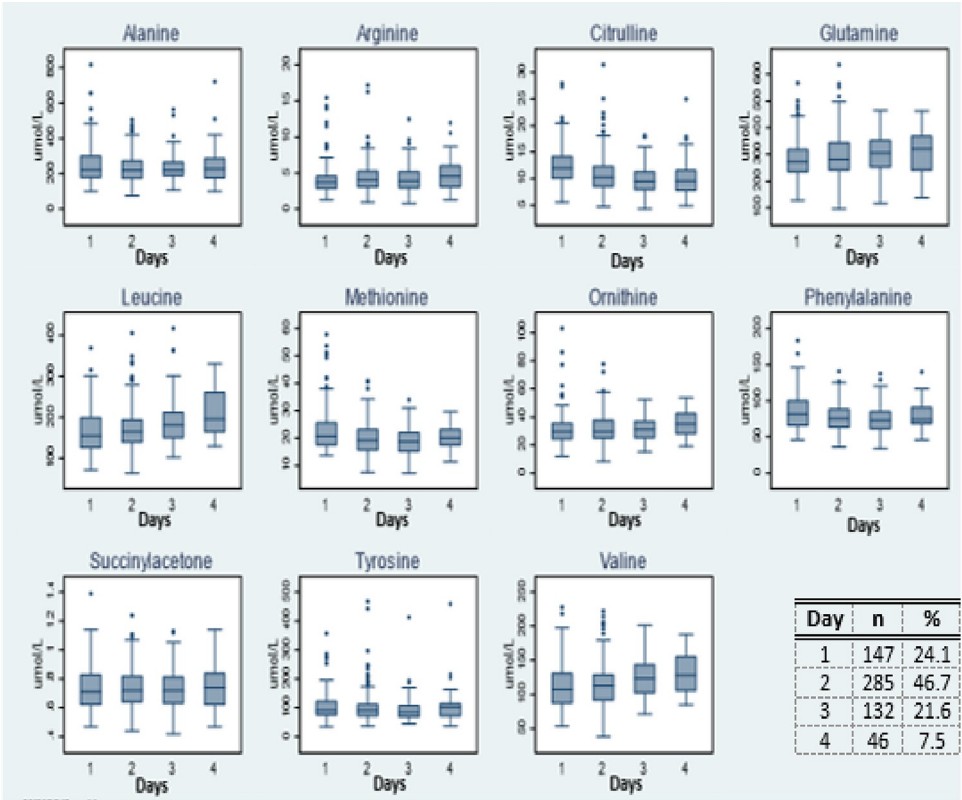

**Fig 2. Age wise distribution of DBS biomarkers according to age in cohort of neonates from AMANHI bio repository in Pakistan (n = 610).** a: Age wise distribution of DBS amino acid and succinylacetone concentrations according in age of neonates from AMANHI bio repository in Pakistan (n = 610). b: Age wise distribution of DBS acylcarnitine concentrations according to age of neonates from AMANHI bio repository in Pakistan (n = 610).

## Discussion

This is the first study from Pakistan, reporting normative metabolic profiles of amino acids, acylcarnitines and succinylacetone. Literature reports that SGA and preterm babies are prone to develop metabolic disorders such as hypertension, diabetes, and cardiovascular diseases in later life [27,28]. The study subjects were categorized in to three groups to evaluate the mean difference due to GA and term birth. Clinically significant effect of GA, weight, gender, and time of sampling on amino acid, succinylacetone and acylcarnitine blood spot concentrations were not identified in the current study. This was contrary to our expectations in comparison to findings from some previous studies [16–18]. Most amino acid concentrations including mean alanine levels in Pakistani newborns from this study were comparable with the published studies of liquid chromatography with tandem mass spectrometry (LC-MS/MS) on DBSs conducted in Thai newborns aged 0–4 days [29]. Wilson et al describe the effects of maturity on arginine, leucine, and valine that were at least 50% different between the cohorts of extremely premature and term children which was not seen in our study population [30]. However, in our study tyrosine levels in premature were higher (succinylacetone being normal) with a mean difference of 27.63 μmol/L. Elevated tyrosine with normal succinylacetone is primarily associated with transient tyrosinemia of the newborn which is common. Literature shows that up to 10% percent of newborns may have transient tyrosinemia, possibly due to vitamin C deficiency or immature hepatic enzymes which is benign and resolves without sequelae [31]. Reese et al studied the effect of GA and chronological age on 15 amino acids and 35

acylcarnitines in 995 infants' blood taken within the first 24 hours after birth and on approximately days 7, 28, and 42 of life [32]. Out of the total, 21% of the neonates from this study had amino acids and acylcarnitines values above the pre-specified cutoffs used to identify infants with IMDs. None of the abnormal values could be explained by contamination of the blood sample, or by inappropriate collection methods [33].

A Turkish group of researchers in a retrospective cohort study observed higher alanine levels in SGA neonates when compared with AGA neonates (p values < 0.05). Differing to the Turkish findings, in our study the DBS alanine levels were higher in AGA (247.71 ±100.32 µmol/L) as compared to SGA newborns (232.16±80.64 µmol/L); *p* value 0.04. Alanine is the key product of lactate metabolism and assists in glucose homeostasis, and higher levels of alanine in preterm neonates have been reported to increase the risk of diseases related to metabolic syndrome in later life [34]. We also noted higher mean alanine levels in full terms as compared to preterm with a mean difference of ±22 µmol/L. In the same Turkish study, methionine, isoleucine levels, C0, C2, C4, C5, C10:1, C18:1, C18:2, C14-OH, and C18:2-OH were higher and C3 and C6-DC levels were lower in SGA when compared with AGA newborns which was not seen in our sample population (p < 0.05) [33]. In another study conducted on full-term newborns (n = 6131) born in Iceland from 2009 to 2012 with newborn screening samples collected 72–96 hours after birth showed that both LBW (n = 36) and extremely macrosomic newborns (n = 37) show dissimilar metabolomic profiles compared to AGA neonates. For LBW the mean differences were higher than for AGA neonates in relation to C0, C8:1, C14:2, C18:2, while lower for C4OH. The greatest difference between LBW and AGA neonates was in levels of C0 (9 µmol/L). They also reported higher alanine in LBW babies contrary to our findings. However, the subjects in their study were only classified according to their weight without taking GA into account [35]. Dietzen DJ et al generated the 2.5th–97.5th percentile distributions of amino acids on DBS of newborns (n = 310) aged 0–4 days analyzed using LC-MS/MS, same instrument as ours [36]. The DBS specimens were obtained from a biobank stored at -80˚C. Comparing results from the current study and those reported by Dietzen DJ et al, RIs for alanine. leucine. phenylalanine and tyrosine, from our study population were slightly higher while arginine and glutamine were lower. No difference in RI of citrulline and methionine were obvious. However, a parallel comparison cannot be done as acylcarnitine and succinylacetone were not included and RIs were not determined taking gender, weight, and GA into account like our current study. Additionally, the differences in metabolites reference ranges between the previous studies and our work could be due to differences in methodology, equipment, reference sample, measurement conditions, or from dietary patterns which were not studied or compared. Further studies need to be conducted to determine the impact of pre-analytical variables like storage conditions, time between collection and freezing, impact of freeze-thaw cycles on amino acids, acylcarnitine and succinyl acetone levels.

The strengths of the study include sampling of early newborns from diverse ethnicities existing in Pakistan. Moreover, DBS collection for the biorepository was done using standardized methods, with all DBS cards stored immediately at -80˚C and shipped for analysis within two months of collection under dry ice with temperature control to avoid transportation errors. We are confident of the quality of DBS cards as these samples were collected for the biorepository, transported, and processed under strict quality control measures and methods that have also been published [20]. A noticeable limitation of the study is the collective aggregate in each age group was inadequate to calculate age specific (0–4 days) RIs in neonates. This community derived RI may not be representative of all Pakistani babies as ours was a convenience sampling and subjects were predominantly from a single peri-urban community; however, all major ethnicities of Pakistan were represented. Besides being the only markers for

identifying aminoacidopathies in newborns, the amino acid levels, acylcarnitne in neonates is also a sign of their metabolic and nutritional status. The amino acids also reflect the nutritional (protein) intake of the mothers before and throughout pregnancy. It would be interesting to examine whether amino acid and acylcarnitine concentrations vary with maternal and neonatal nutritional status and intrauterine growth outcomes. The intrauterine environment can influence the metabolome during gestation, the results of which often manifest in the metabolic status of infants in later life.

## Conclusion

Our study reflects the metabolic profile of newborns in Pakistani community. By examining reference data from six hundred and ten newborns, this study established RIs for DBS amino acids, acylcarnitines and succinylacetone that can be applied for screening of some neonatal genetic metabolic diseases using tandem mass spectrometry in Pakistan irrespective of gender, GA and weight.

## Supporting information

**S1 Table. Reference intervals of amino acids, succinylacetone and acylcarnitine in neonatal dried blood spots.** Appropriate for gestational age (AGA) means the baby's weight is appropriate for the GA (weight between 10th and 90th percentile). Small for gestational age (SGA) means a baby's weight is less than expected for the GA (weight less than 10th percentile). *p-values are calculated using two sample t-test for gender difference.
(DOCX)

## Acknowledgments

We would like to acknowledge collaborators in Iowa University, US where analysis was performed.

## Author Contributions

**Conceptualization:** Imran Nisar, Fyezah Jehan.

**Data curation:** Lena Jafri, Aysha Habib Khan, Fyezah Jehan.

**Formal analysis:** Lena Jafri, Aysha Habib Khan, Muhammad Ilyas, Javairia Khalid.

**Funding acquisition:** Imran Nisar, Javairia Khalid, Aneeta Hotwani, Fyezah Jehan.

**Investigation:** Lena Jafri, Imran Nisar, Javairia Khalid, Fyezah Jehan.

**Methodology:** Lena Jafri, Aysha Habib Khan, Imran Nisar, Hafsa Majid, Fyezah Jehan.

**Project administration:** Lena Jafri, Imran Nisar, Javairia Khalid, Aneeta Hotwani, Fyezah Jehan.

**Resources:** Aneeta Hotwani, Fyezah Jehan.

**Software:** Lena Jafri, Muhammad Ilyas, Fyezah Jehan.

**Supervision:** Lena Jafri, Aneeta Hotwani, Fyezah Jehan.

**Validation:** Lena Jafri, Aysha Habib Khan, Muhammad Ilyas, Hafsa Majid.

**Writing – original draft:** Lena Jafri.

**Writing – review & editing:** Lena Jafri, Aysha Habib Khan, Muhammad Ilyas, Imran Nisar, Hafsa Majid, Fyezah Jehan.

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
