## [Decision Letter · Decision Letter 0]

15 Aug 2022

PONE-D-22-19104Metabolomics of a neonatal cohort from the Alliance for Maternal and Newborn Health Improvement Biorepository: Effect of preanalytical variables on reference intervalsPLOS ONE

Dear Dr. Jafri,

Thank you for submitting your manuscript to PLOS ONE. After careful consideration, we feel that it has merit but does not fully meet PLOS ONE’s publication criteria as it currently stands. Therefore, we invite you to submit a revised version of the manuscript that addresses the points raised during the review process.

We look forward to receiving your revised manuscript.

Kind regards,

Iman Al-Saleh

Academic Editor

PLOS ONE

Journal Requirements:

a) Did participants provide their written or verbal informed consent to participate in this study?

**Reviewers' comments:**

Reviewer's Responses to Questions

**Comments to the Author**

1. Is the manuscript technically sound, and do the data support the conclusions?

Reviewer #1: Partly

Reviewer #2: Yes

Reviewer #3: Yes

2. Has the statistical analysis been performed appropriately and rigorously? 

Reviewer #1: Yes

Reviewer #2: Yes

Reviewer #3: Yes

3. Have the authors made all data underlying the findings in their manuscript fully available?

Reviewer #1: Yes

Reviewer #2: No

Reviewer #3: No

4. Is the manuscript presented in an intelligible fashion and written in standard English?

Reviewer #1: Yes

Reviewer #2: Yes

Reviewer #3: Yes

5. Review Comments to the Author

**Reviewer** #**1**: The objective of this study was to determine reference intervals (RI) for dried blood spot measurements of amino acids, acylcarnitines, and succinylacetone of neonates in Pakistan. The authors examined whether mean levels of the analytes differed statistically by gender, term, or gestational age. Although some statistically significant differences were observed, the authors concluded that none of these were clinically significant and, thus, did not recommend separate RI for different subgroups. These are important data to support development of a national newborn screening program.

Comments:

Line 150: “171 (53%) were from females”. According to Table 1, 325 samples were from females.

Lines 153-155: “145 (23.7%) neonates being low birth weight (LBW), 141 (23.1%) were SGA and 87 (14.3%) being preterm.” The percentages for normal weight and low birth weight in table 1 appear to be incorrect (e.g., 465/610 is 76.2%, not 75.5%; 145/610 = 23.8%, not 24.5%). Please define normal weight and low birth weight in Methods. Please add data for SGA and AGA to Table 1.

When describing the table 2 results, the authors state that although there are statistically significant differences in some of the analytes between males and females, the differences did not seem clinically significant and thus did not justify creation of separate RI. The process for weighing clinical vs. statistical significance is not clear for values with more than negligible differences (e.g., the 97.5% limit for C2 is 47.08 in males vs. 41.48 in females). Perhaps the authors could add to Methods the process for making these judgements. For example, were all statistically significant differences reviewed by one author or several authors who came to a consensus about clinical significance? Was literature consulted?

Some authors have proposed more objective criteria to determine the need for partitioning (e.g., PMID 11805016). Did the authors consider a similar approach and decide against it? Perhaps the authors could note in Methods or the Discussion why they did not use such an approach.

Consider expanding table 2 to show the mean and median for males and females separately.

Figure 1 footnote: “No statistically significant difference (p value >0.05) was found amongst gender in the amino acids, succinylacetone and acylcarnitine concentrations studied.” This is not accurate as some statistically significant differences were observed for acylcarnitines according to table 2.

In Line 154 the authors wrote that 141 were SGA, but table 3 shows 219 are SGA. Please clarify.

In table 3, testing differences in means does not seem to be the correct focus for establishing RI. A mean value can be statistically significantly different between groups while the RI are very similar, as was shown in table 2.

In the footnote of table 3, the authors mention accounting for false discovery rate. Details about this approach and when it was applied should be added to Methods.

For the analyses in figure 2, please report the number of samples collected on each day (i.e., the n for each group). Also, why is day 0 shown in 2a, but not 2b? Also, please clarify in the figure that the x-axis is showing days.

For the Dietzen study, I am curious if they also stored DBS cards immediately at -80 degrees and if they used a similar assay to measure the metabolites. From lines 276-278, it is not clear if the authors are saying the Dietzen study did not report these details at all in their study or if Dietzen used different methods than theirs so the results cannot be compared? If they did report their methodology, this would be useful to describe as that might suggest that further work needs to be done to determine the impact of storage conditions (e.g., time between collection and freezing, freezer temperature) or other pre-analytic variables on metabolite levels.

**Reviewer** #**2**: Jafri et al. present a relevant and sound research article establishing reference intervals of diverse metabolites for newborn screening in Pakistan. Scientific and statistical methods applied are sound and the study cohort is well described. Findings support conclusions and the manuscript is well written. I only have a minor comment, for Tables 1 and 2 please add a description of what p-values identify (which test statistic was used, which differences were measured) to improve readability.

**Reviewer #**3: This article describes the results metabolite analysis of dried bloodspots in 610 neonates. DBS analysis is critical to modern neonatology and is now mandatory in the U.S. and many other countries. However, it has been shown that these assays are sensitive to environmental factors. This work, therefore, is of critical importance in establishing an appropriate baseline for deployment of DBS metabolite screening globally. The study is designed and executed well, and the article is easy to follow. My comments for improving the manuscript are provided below:

1) The authors have demonstrated that the metabolites are not associated with GA, weight, gender, and time of sampling. However, it may be the case that a multivariable model can combine multiple weak associations to identify a signature strong associated with these factors. Adding a simple multivariable linear model to the analysis will be very valuable.

2) It has been shown that hypertensive disorders of pregnancy and diabetes can have a profound affect on metabolites measured in DBS. It would be great if the authors could add hypertension, preeclampsia, and diabetes to the analysis (if enough patients with these phenotypes are available).

3) It is unclear to me why this data cannot be shared. PHI information, of course, would require institutional review and approval. But metabolites themselves and associated demographic/phenotypical information is not PHI and is regularly shared publicly. This will increase the impact of the article as others will be able to reuse the data. This is, of course, subject to institutional and governmental regulations in various countries that I may not be aware of.

Taken together, I believe this is an important article and all my comments can be addressed in a straight-forward revision. I would, therefore, encourage the editorial board to strongly consider the publication of this article.

Reviewed by Nima Aghaeepour

(Voluntary disclosure of my name)

6. PLOS authors have the option to publish the peer review history of their article (what does this mean?). If published, this will include your full peer review and any attached files.

Reviewer #1: No

Reviewer #2: No

Reviewer #3: No

---

## [Author Response · Author response to Decision Letter 0]

14 Dec 2022

Dear Editor,

We have revised the manuscript as per reviewers and editorial comments and suggestions. We have submitted a revised version of the manuscript (marked and unmarked copies) and addressed all comments point by point in the rebuttal letter attached.

Regards

Lena

---

## [Editor Report · Decision Letter 1]

19 Dec 2022

Metabolomics of a neonatal cohort from the Alliance for Maternal and Newborn Health Improvement Biorepository: Effect of preanalytical variables on reference intervals

PONE-D-22-19104R1

Dear Dr. Jafri,

We’re pleased to inform you that your manuscript has been judged scientifically suitable for publication and will be formally accepted for publication once it meets all outstanding technical requirements.

Kind regards,

Iman Al-Saleh

Academic Editor

PLOS ONE

Additional Editor Comments (optional):

The authors addressed all the reviewers' comments satisfactorily.

---

## [Editor Report · Acceptance letter]

27 Dec 2022

PONE-D-22-19104R1 

Metabolomics of a neonatal cohort from the Alliance for Maternal and Newborn Health Improvement Biorepository: Effect of preanalytical variables on reference intervals 

Dear Dr. Jafri:

I'm pleased to inform you that your manuscript has been deemed suitable for publication in PLOS ONE. Congratulations! Your manuscript is now with our production department. 

Kind regards, 

on behalf of

Dr. Iman Al-Saleh 

Academic Editor

PLOS ONE